# Improvement of Therapeutic Value of Quercetin with Chitosan Nanoparticle Delivery Systems and Potential Applications

**DOI:** 10.3390/ijms24043293

**Published:** 2023-02-07

**Authors:** Michael Kenneth Lawson

**Affiliations:** Department of Galenic Pharmacy, Comenius University Bratislava, Odbojárov 10, 83232 Bratislava, Slovakia; lawson2@uniba.sk

**Keywords:** quercetin, chitosan, nanoparticles, drug delivery system, cancer treatment

## Abstract

This paper reviews recent studies investigating chitosan nanoparticles as drug delivery systems for quercetin. The therapeutic properties of quercetin include antioxidant, antibacterial and anti-cancer potential, but its therapeutic value is limited by its hydrophobic nature, low bioavailability and fast metabolism. Quercetin may also act synergistically with other stronger drugs for specific disease states. The encapsulation of quercetin in nanoparticles may increase its therapeutic value. Chitosan nanoparticles are a popular candidate in preliminary research, but the complex nature of chitosan makes standardisation difficult. Recent studies have used in-vitro, and in-vivo experiments to study the delivery of quercetin alone or in combination with another active pharmaceutical ingredient encapsulated in chitosan nanoparticles. These studies were compared with the administration of non-encapsulated quercetin formulation. Results suggest that encapsulated nanoparticle formulations are better. In-vivo or animal models simulated the type of disease required to be treated. The types of diseases were breast, lung, liver and colon cancers, mechanical and UVB-induced skin damage, cataracts and general oxidative stress. The reviewed studies included various routes of administration: oral, intravenous and transdermal routes. Although toxicity tests were often included, it is believed that the toxicity of loaded nanoparticles needs to be further researched, especially when not orally administered.

## 1. Introduction

Nanoparticles (NPs) and microparticles (MPs) are revolutionizing technology. Such particles may be made of many materials, from inorganic compounds to biologically derived polymers.

One particularly popular material is chitosan. Chitosan nano- and microparticles have found many useful applications, from the dermal delivery of sun-protecting substances [1] to the removal of industrial pollution [2,3,4,5]. It also has intrinsic antimicrobial properties [6]. Recent reviews about chitosan have already been written, for example, by Mikušová and Mikuš [7].

This review is restricted to considering nanoparticles containing chitosan as a drug delivery system carrying the active pharmaceutical ingredient (API) quercetin and, perhaps, other APIs. Quercetin is interesting because it is a common phytochemical with many possible therapeutic effects, for example, against cancer and diabetes. It is difficult to deliver with conventional drug delivery systems, but it is suggested that its therapeutic value can be increased with better drug delivery systems and administered to more specific sites such as the skin. It may also act synergistically with other APIs, in particular with anti-cancer drugs.

This review shows the recent research done for a variety of medical applications, from treating various types of cancer to sun protection, wound healing and protection from general oxidative stress. The papers reviewed illustrate the methods for the characterisation of nanoparticles and toxicity tests. There are also sections reviewing the properties of quercetin and chitosan.

## 2. Quercetin

### 2.1. Basic Properties of Quercetin

Quercetin is a member of a group of related compounds known as flavonoids. It is a secondary metabolite in plants. It is believed, along with carotenoids, to protect plant tissue against UV-light-induced oxidation [8]. Quercetin is a relatively small (molar mass 302.24 g mol^−1^) hydrophobic molecule (octanol–water partition coefficient 1.82 ± 0.32) [9]. Its structure is given in Figure 1. While the non-polar groups of the skeleton give it its hydrophobic nature, it also contains polar hydroxyl groups. It is the least lipophilic compared with other flavonoids [9]. For humans, it is an important dietary nutrient, an antioxidant and a potential therapeutic agent.

### 2.2. Derivatives and Metabolites of Quercetin

Actually, quercetin exists as several derivatives in plants. These derivatives are usually quercetin bound to sugars and alcohols [10]. These derivatives are less hydrophobic than the unsubstituted form of quercetin [9]. Furthermore, some derivatives may be more therapeutic than others. For example, onions contain quercetin glucoside, while apples contain quercetin rhamnoside and quercetin galactoside. It has been shown that quercetin glucoside has higher bioavailability than the derivatives found in apples [11]. The most common forms of quercetin ingested orally are quercetin 3-O-glucoside, quercetin 4′-O-glucoside, quercetin 3,4′-O-diglucoside, and the unsubstituted quercetin known as quercetin aglucone [12].

Quercetin is mainly absorbed in the intestine by membrane-related transporters such as sodium-dependant glucose co-transporters (SGLTs) and organic anion transport polypeptides (OATPs). Then, there is the phase II metabolism (that is, conversion to less lipophilic, more water-soluble compounds easier to eliminate) in the intestine and liver. This gives the methylated, sulphated, and glucuronidated forms in the circulatory system. There is eventual excretion via bile, urine and faeces [12,13].

Quercetin not absorbed by the intestine reaches the colon. Here, there is a breakdown by intestinal microbiota to give phenolic acids and aromatic compounds. The most common are 3,4-dihydroxyphenylacetic acid, (3-hydroxyphenyl) propionic acid, 3,4-dihydroxybenzoic acid, and 4-hydroxybenzoic acid. These are absorbed by colonocytes. There is then the phase II metabolism, producing 3-hydroxyphenylacetic acid, 3,4-dihydroxyphenylacetic acid, and 4-hydroxy-3-methoxyphenylacetic acid. The final products reach the circulatory system and are excreted into the urine [12].

In or before reaching the liver, there are the phase I and II metabolisms of quercetin and its metabolites. Phase I includes oxidation, reduction and hydrolysis. Phase II metabolism involves glucuronidation, sulfation and methylation, aiding excretion through bile and urine. The various xenobiotics are uptaken in the liver by various kinds of transporters [12].

Some quercetin metabolites, such as *p*-hydroxyphenylacetic acid and 3,4-dihydroxyphenylacetic acid, have been shown to have positive health effects [12,14]. Additionally, 3-hydroxyphenylacetic acid, 3,4-dihydroxyphenylacetic acid and 4-hydroxy-3-methoxyphenylacetic acid, from the colon microbiota, reach certain tissues and have more beneficial effects than quercetin itself [12,15,16].

Hai et al. describe how quercetin and quercetin metabolites may help in promoting intestinal health as well as protecting the liver and kidneys from oxidative stress, inflammation and fibrosis [12]. It is also believed that quercetin metabolites have a positive effect on the microflora in the intestine [12,17].

It has been found that in blood plasma, quercetin itself is not detected using HPLC (high-pressure liquid chromatography) but that quercetin 3-O-β-D-glucuronide and quercetin-3′-sulphate are the predominant derivatives detected [18,19]. The structures of these derivatives of quercetin are shown in Figure 2.

### 2.3. Beneficial Effects of Quercetin on Health

Quercetin has been found to have many beneficial effects when consumed by animals. It is a very good antioxidant. Some of the more well-known reasons for its beneficial effects as an antioxidant are that it:Chelates metal ions, preventing the metal ions from catalyzing oxidation reactions;Inhibits lipid peroxidation by donating hydrogen ions to α-tocopherol radicals, regenerating this primary oxidant;Elevates the level of glutathione (another important molecule in reducing oxidative stress).

The beneficial properties and therapeutic potential of quercetin are many. Firstly, as an antioxidant, it is the most effective free radical scavenger in the flavonoid family [20]. It is considered an even more powerful oxidant than vitamins C and E [21]. In-vitro antioxidant mechanisms include the direct scavenging of free radicals [22,23], the chelation of Cu(II) and Fe(II) metal ions [24,25] and the inhibition of lipid peroxidation [26,27]. In-vivo mechanisms include the regulation of glutathione [28], the increased expression of some antioxidant enzymes [29] and a beneficial influence on signal transduction pathways related to oxidative stress [30,31].

Quercetin also has antimicrobial [32] and anti-tumour properties [33], anti-inflammatory and immunosuppressive effects [34,35,36], and cardiovascular protection [37]; it can relieve mycotoxin toxicity [38] and has the potential to treat diabetes [39,40,41], inhibit fat production [42], and reverse cognitive impairment [43]. The mechanisms include the modification of signalling pathways and the inhibition of enzymes.

### 2.4. Effect of Quercetin on Signalling Pathways

Many therapeutic agents act by modulating signalling pathways, and this is especially true for new types of cancer control drugs such as ibrutinib [44], where the signalling events of cancer cells are abnormal and undesirable. Quercetin can also modulate signalling pathways. The signalling pathways may be that of the patient or microorganisms that harm the patient. Quercetin may disrupt the signalling pathways of microorganisms, inhibiting their growth or inhibiting cancer cells from dividing or causing cell apoptosis when and where appropriate. Quercetin can also inhibit certain enzymes, such as tyrosinase. A list of how quercetin can influence signal molecules and enzymes in a way that treats certain conditions is given in Table 1.

For example, quercetin can upregulate p53 and downregulate human epidermal growth factor receptor 2, giving it anti-cancer potential, and it can reduce infection by microorganisms by inhibiting prokaryotic DNA gyrase [45]. It can help treat diabetes by inhibiting certain enzymes, for example, α-glucosidase and α-amylase [40]. It can inhibit tyrosinase and hence can be used as a skin whitener in cosmetics [46]. More mechanisms for these beneficial effects are given by Salehi et al. [46].

### 2.5. Contraindications, Possible Toxicity and Dangers of Quercetin Supplements

With so many papers emphasizing the positive health benefits of quercetin, it is important to consider the possibility of toxicity, negative health effects and possible contraindications of quercetin with other APIs, excipients and diet. Quercetin is considered a food rather than a medical product and so is largely unregulated as a supplement, with no premarket safety and efficiency testing and no warning label to warn of potential adverse effects and drug interactions [47]. Quercetin supplements can be sold in very high doses (up to 1000 mg/unit), and 4000 mg/day is recommended [47].

The toxicity of quercetin has been explored [48,49,50,51]. Generally, it is not considered toxic, but this only holds if it is taken orally in a non-encapsulated formulation. In-vitro studies suggest mutagenic properties, but in vivo studies do not [51]. The toxic effects of nanoparticles loaded with quercetin are another matter, and this needs to be tested for each type of nanoparticle system. Nan et al. used in-vitro cultivated human skin cells, HaCaT, and the MTT (3-(4,5-dimethylthiazol-2-yl)-2,5-diphenyltetrazolium bromide) assay to test the toxicity of quercetin and quercetin in chitosan-TPP nanoparticles and found that the nanoparticle carrying quercetin was less toxic than free quercetin at concentrations from 10 to 40 µg/mL [1].

A literature search for ‘contraindications of quercetin’ did not give recent scientific articles on this subject. This may be the reason why so many articles are about the benefits of quercetin, with little evidence reported for the dangers of quercetin. Several health articles, however, warn against the dangers of taking high-dose quercetin supplements. These health articles usually include a list of scientific articles that the article has used.

According to one health article, quercetin should not be taken with the following:Antibiotics (quercetin may reduce effectiveness);Anticoagulants (quercetin may enhance the effect of these blood thinners);Chemotherapy (may increase the effect of doxorubicin or decrease the effect of cisplatin—doctors in disagreement as to whether antioxidants are good or bad during chemotherapy);Corticosteroids (quercetin may prolong the circulation time of these drugs);Cyclosporine (quercetin may interfere with absorption);Digoxin (quercetin may increase the risks of this API);Fluoroquinolones (quercetin may reduce the effectiveness of API);Medications changed by the liver (quercetin may alter the metabolism of APIs by the liver) [52].

The article also states that doses greater than 1 g per day can damage kidneys, and the possible side effects of quercetin are headache and upset stomach [52]. The article cites research papers, although many of the papers are about the benefits of quercetin, with very few examining the negative health effects of quercetin.

Another article states that quercetin is contraindicative with sophretin or meletin, has mild interactions with alvimopan, armodafinil, ciprofloxacin, fexofenadine, fleroxacin, characterize, levofloxacin, loratadine, moxifloxacin, norfloxacin and ofloxacin and more serious interactions with everolimus and topotecan [53]. However, the original source of this information is not clear.

Regarding scientific journals, the following two studies summarised below are often cited regarding toxicity issues:

Boots et al. have published a paper which investigates the possible in-vivo toxicity of quercetin using rat lung epithelial (RLE) cells. They point out that the main oxidation product of quercetin has high reactivity with thiols, which can lead to the loss of protein function. Their studies showed a depletion of GSH (glutathione), increased leakage in LDH (lactate dehydrogenase), and an increase in cytosolic free calcium concentration [54].

Harwood et al. published a critical review that recognises that in-vitro studies suggest some toxicity while in-vivo studies do not, concluding that quercetin as an additive in food is probably safe. However, they propose that the reason for non-toxicity in-vivo is probably the gastric and first-pass metabolism of quercetin [51]. Thus, formulations protecting quercetin from gastric processing and other routes of administration need to address the possibility of carcinogenetic, mutagenic or other toxic effects of quercetin.

In addition to the above two articles, Vida et al. quote several relevant studies during their investigations of online quercetin supplements. They cite a number of articles that suggest some possible interactions with drugs [47]. Poór et al. reported that quercetin interacts with serum albumin and suggested this may compete with albumin-transported drugs [55,56,57]. Kimara et al. reported inhibitory effects on cytochrome P450 enzymes [58], and Wang et al. reported effects on P-glycoprotein transport ability [59]. Andres et al. suggested warning patients taking chemotherapeutic or cardiovascular medications with a narrow therapeutic index and medications which have a narrow therapeutic window, such as warfarin and digoxin [60]. Additionally, metabolites of quercetin (e.g., isorhamnetin, quercetin-3′-O-sulphate, quercetin glucuronides) affect enzymes and transporters, including CYP2C9, CYP3A4, OATs and GLUT (a glucose transporter) [56,57,58,61,62].

To conclude, the possible harmful effects of quercetin have not been so well investigated due to its status as a naturally occurring ingredient of food.

### 2.6. Difficulties in Quercetin Administration as a Therapeutic API

Quercetin itself is difficult to deliver because of low aqueous solubility (1 µg/mL or 3.31 nanomoles/mL), low bioavailability, poor absorption (<10%), a high metabolic rate (<1 h), rapid clearance from the body, inactive metabolic products and instability [11,63,64]. Derivatives, particularly those containing a sugar moiety, can improve the aqueous solubility to a limited extent. Oral administration has a major disadvantage in that there is rapid gastrointestinal digestion and many of the sugar moieties are removed, reducing solubility [19]. On the other hand, this may be the reason that quercetin has been largely found to present no toxicity to the consumer [51].

Better formulations that will improve the bioavailability and stability of quercetin may depend on the route of administration, which will depend on the therapeutic purpose and target of quercetin. For example, if quercetin is used as an antioxidant to protect the skin, then a dermal route will be preferred and the formulation should have rheological properties that make it easy to spread on the skin. If the purpose is to treat neurological degenerative diseases, then a nasal route may be the preferred route of administration, with good interaction with mucus. Alternatives to oral administration, for example, dermal administration, may also increase bioavailability, avoiding gastric digestion and first-pass metabolism. Numerous approaches have been undertaken, involving the use of promising drug delivery systems such as inclusion complexes, liposomes, nanoparticles or micelles [63]. In this paper, we will limit the formulations to nanoparticles, specifically chitosan nanoparticles.

## 3. Dosage Forms and Means of Administration

It is important that an active pharmaceutical ingredient (API), commonly called a drug, is mixed with other substances, called excipients, in a specific mixture so that the final product, called a medicine, can be given to the patient in a convenient form and the API has the optimal therapeutic effect.

Excipients may have many purposes, including helping with the absorption or adsorption of the API, adjustment of the viscosity, improvement of stability during storage of the medicine, adjustment of organoleptic properties, increase in bulk or weight, masking an unpleasant taste and giving a pleasant appearance or colour. Excipients may be classified according to their function, for example, preservatives, antioxidants, fillers and colourants.

The medicine can be in the form of a solid tablet that is swallowed, a spray that can be sprayed up the nasal cavity, a liquid that can be swallowed, or a cream that can be applied to the skin. These so-called dosage forms can be classified as solid (powders, granules, tablets), semisolid (gels, ointments, creams, pastes) or liquid (solutions, syrups, emulsions, suspensions). Sprays and aerosols are usually suspensions, solid or liquid particles, in a gas rather than a liquid.

The dosage form chosen depends on several considerations, which include the API, the disease to be treated, the target of treatment, and the route of administration. Routes of administration include oral, intravenous, intramuscular, subcutaneous, intraocular, pulmonary, transdermal, nasal, rectal, vaginal, sublingual/buccal or applied directly at the site of disease, for example, in the ear.

Quercetin has potential therapeutic value for a number of disease states, but the physicochemical properties may exclude certain dosage forms and routes of administration. While there are many potential dosage forms, it seems, to the author’s knowledge, that quercetin is presently mainly available as a supplement in the form of a tablet, capsule or powder that is administered orally [47,65] or as a cream for application to the skin as a cosmetic [66] or to treat itching skin [67].

As previously mentioned, the main limitation of quercetin is its low bioavailability. Oral administration has the added disadvantage of the rapid gastrointestinal digestion of quercetin [64]. Thus, the present dosage form of quercetin supplements is far from ideal. Apart from the transdermal route, other traditional dosage forms and routes of administration seem not to have been well explored. However, new innovative dosage forms are being well explored and developed to overcome the limitations. One of the most common and powerful new innovative dosage forms is to encapsulate the API (i.e., quercetin) in a nanoparticle. The API-loaded nanoparticles can be carried in a more conventional dosage form, such as a gel, cream or solution.

With conventional dosage formulations, many potential APIs may suffer from limitations such as low bioavailability, long-term stability, toxic effects at low doses, poor targeting, degradation by enzymes and the immune system, excretion, etc. Improved innovative delivery systems, such as nanoparticle delivery systems, can greatly widen the number of APIs on the market. They protect the API from being enzymatically metabolized, increasing stability and circulation time [68,69], and can also have specific targeting moieties attached to the surface, which improve targeting and thus reduce toxicity to other parts of the organism.

To conclude this section, while quercetin supplements are available in conventional dosage forms (usually taken orally), these dosage forms do not adequately address the problem of low bioavailability, and the means of administration is limited by the physicochemical properties of quercetin. They are also not designed to target specific disease states and sites. To overcome these limitations and improve the therapeutic value and specific target of quercetin, new innovative dosage forms are being developed, the most common being to encapsulate the API (i.e., quercetin) in a nanoparticle.

## 4. Chitosan and Chitosan Nanoparticles

In the last section, it was concluded that a better means of delivering quercetin is to encapsulate it in a nanoparticle. There are many types of nanoparticles made of many types of material. In this review, only nanoparticles composed of chitosan alone or in combination with other materials are considered.

Why chitosan? Other materials used for nanoparticles include silica, PLGA (polyethylene glycol), PLA (polylactic acid) and natural or synthetic phospholipids in liposomes. These have been briefly discussed by Nam et al. [64]. Some disadvantages or problems that are difficult to overcome include the use of organic solvents in making PLGA and PLA nanoparticles, low drug loading, less stability and the high cost of liposomes.

### 4.1. Chitosan

Chitosan is a linear polymer of two monomers, glucosamine and acetylglucosamine. It is produced from chitin, which contains much lower amounts of glucosamine. Chitin is usually from crab or shrimp exoskeletons and converted to chitosan by partial deacetylation [7]. The deacetylation of chitin to produce chitosan is usually achieved by hydrolysis of the acetamide groups with concentrated NaOH or KOH (40–50%) at temperatures above 100 °C, generally under heterogeneous conditions [70]. The polymer can vary in average molecular weight from 300 to 1000 kD and deacetylation degree (DDA) [71]. Unlike most industrial high molecular weight polymers, it can be protonated to form a cationic polyelectrolyte [72]. When positively charged, it can attach to the negatively charged mucosal membranes or carry negatively charged APIs, including nucleic acids. Chitosan is claimed to be GRAS (Generally Recognised As Safe) and bioabsorbable [72]. It is of low toxicity because of its structural similarity to natural glycoaminoglycans and is biodegraded into harmless amino sugars that are absorbed completely by the body [7].

Chitosan can be characterized by its average molecular weight (Mw), distribution of molecular weights, DDA and pKa. Usually, the pKa of chitosan is about 6.5, but there is low protonation in neutral solutions [73]. A larger charge on the chitosan polymer can increase its carrying ability and can serve as a pH-dependent drug carrier. As already mentioned, there are variations in average molecular weight from 300 to 1000 kD and degree of deacetylation from 50% to 90%, depending on the source and preparation procedure [58,74]. Ideally, all these parameters need to be specified when comparing different studies. This specification is a simplification because the parameters do not uniquely characterize the polymer. The distribution of the two copolymers, glucosamine and acetylglucoseamine, along the chain may vary for different sources and may not necessarily be distributed in a random way according to the Markovian statistical model. Furthermore, some methods of determination of average Mw or DDA can be inaccurate or problematic [72].

For biological applications, chitosan has the following advantageous properties:It is biodegradable and biocompatible [75].It is soluble in weak acids, such as acetic acid or lactic acid, at pH 6.5.It can interact with some important anionic pharmaceutically active products, for example, nucleic acids.It can easily form nanoparticles by ionic interaction with negatively charged substances such as tripolyphosphate (ionic gelation method) [76].Nanoparticles can be made under mild conditions with no organic solvents [7].It can bind to negatively charged substances in the body, such as mucosal membranes, thus prolonging the time of contact, enabling any carried pharmaceutical time for penetration [77,78].Its carrying ability increases with charge, which can be changed with pH [79].It has a chelating ability, which could inactivate free-radical-producing metal ions.Chitosan can be modified to optimise beneficial properties according to the needs of the application, for example, giving better aqueous solubility. This can be done by changing the average molecular weight, degree of deacetylation, cross-linking, adding covalently bound functional groups and moieties, coordinated anions or polyanions, etc. [7].

Furthermore, chitosan has been found to have many other potentially beneficial biological properties. As well as being anti-inflammatory, antioxidant and antimicrobial [80,81], it is also hemostatic, fungistatic, bacteriostatic, spermicidal, anticholesteric and anticancerogenic [79,82,83,84,85,86,87,88,89,90]. It works as an efflux pump inhibitor, thus working against drug resistance [79]. It is also believed that the positive charge causes the reorganisation of cell membrane proteins, causing the tight junctions to open [77].

The limitations of unmodified chitosan are high hydrophilicity, low ductility, a high degree of swelling and insolubility at physiological pH 7.4. [7,86]. Chitosan can be modified not only with respect to molecular weight, DDA and cross-linking but also with the addition of covalently bonded functional groups and moieties. In this way, the desirable properties can be optimised and the undesirable ones minimised, and the modified chitosan fits a specific application. Modifications can occur with a reaction with –NH_2_ of glucosamine or with a –OH group. Added groups may be hydrophobic (alkyl or acrylates), amphiphilic (e.g., cholic or deoxycholic acid), ionic (e.g., sulphates, phosphates, quarternary ammonium), polymers such as PEG (polyethylene glycol) or derivatives with specific substituents such as sugars, cyclic molecules and thiols. Additionally, there is the possibility of linking chitosan molecules together via so-called cross-linkers such as glutaraldehyde, TPP (tripolyphosphate) and EDTA (ethylenediaminetetraacetic acid). Cross-linking molecules are particularly relevant as cross-linking can be used to make chitosan nanoparticles. Such modifications can alter solubility (by addition of alkyl groups), give an anionic character (by sulfonation) and cause less swelling and better mechanical strength (more cross-linking).

Szymańska and Winnicka have reviewed conditions that improve the stability of chitosan. This includes controlling processing conditions such as temperature and humidity, introducing a stabilising compound such as mannitol, sorbitol or glycerol, blending with another polymer such as poly(vinyl alcohol) or poly(ethylene oxide), cross-linking with a cross-linker such as genipin as well as modifying the structure of chitosan, for example, modifying it to chitosanβ-glycerophosphate [91].

The effects of some of these modifications are described below:Alkyl groups, such as carboxymethyl groups, can dramatically increase solubility [92]. Alkylation with glycidicyl trimethyl ammonium chloride gives water solubility at all pHs [93].Sulfonation can turn cationic chitosan anionic. This can make the modified chitosan have water-soluble qualities, better paste fluidity, a high water-reducing ratio, and anticoagulant properties [94].An increase in cross-linking can be achieved with anions, dextran, sulphate, glyoxal, genipin, tripolyphosphate, formaldehyde and glutaraldehyde. More or stronger cross-linking gives mechanical strength to particles, slows down drug release and prevents burst release [86]. Highly cross-linked particles also show less swelling, less inside water penetration and less outside drug diffusion [7].In-situ gelling properties can be improved by thiolation [77].Better mucoadhesion can be achieved by the trimethylation of primary amino groups and PEG-ylation. Thiol groups added to chitosan can interact with the cysteine-rich region of mucous glycol protein. Mucoadhesion can also be improved by the formation of complexes with multivalent drugs, excipients and multivalent inorganic ions [7,77,79].Trimethyl chitosan is soluble at all pHs [7].Trimethyl chitosan enhances permeation by opening tight junctions and increasing paracellular transport [79].Modifications can come at the cost of reproducibility [72].

### 4.2. Chitosan Nanoparticles

Nanoparticles have sizes of around 50–500 nm. Microparticles have sizes in the range of 1–1000 µm. This review is mainly about nanoparticles because they are more easily produced by self-assembly, and their size gives special properties, such as a very large surface area to volume ratio. Nanoparticles can be administered intravenously, whereas microparticles cannot because they would cause the obstruction of blood vessels. On the other hand, microparticles would be better for pulmonary administration, so it is worth considering the use of microparticles for pulmonary administration [95].

It should be borne in mind that the properties of chitosan nanoparticles can be different from the properties of chitosan as a material [96]. Furthermore, chitosan nanoparticles loaded with a drug, in this case, quercetin, may also have modified properties to unloaded chitosan nanoparticles. Jesus et al. also considered undesirable immune responses and unfavourable hemocompatibility and found that chitosan nanoparticles were more cytotoxic than chitosan itself [96].

Microparticles can be made using extrusion, spray or emulsion methods [72]. Chitosan nanoparticles can be made in many ways and a comprehensive list and descriptions can be found in a review by Mikušová and Mikuš [7]. The methods can be classified into 7 groups, namely, covalent cross-linking, self-assembly, emulsion, reverse micellar, drying, precipitation/coacervation and microfluidic [7]. The most common method is by self-assembly, where a negatively charged species, usually a polymer, is attracted to the positively charged chitosan and the two species are bound by electrostatic forces. The chitosan is cross-linked by the negatively charged species to form a chitosan nanoparticle. This method of ionic cross-linking is also called the ionic gelation method. It is especially useful because it occurs under such mild conditions; no heating and no toxic organic solvents are required. The most common negatively charged species used in ionic gelation is the anion of sodium tripolyphosphate (TPP). It is also possible to add a cross-linker that forms covalent bonds with chitosan. Ideally, the cross-linker should also be non-toxic. Examples of these cross-linkers are glutaraldehyde, EDTA and genipin. Alternatives to glutaraldehyde are being sought because it is toxic [97,98]. Covalent cross-linking also has disadvantages, such as lack of swelling, absence of pH release and possible covalent binding to API [72]. It also has advantages such as slowing down drug release and preventing burst release [71].

Nanoparticle size is important as it can affect the absorption, distribution, target size accumulation and elimination of nanoparticles from the body [99]. Nanoparticles can be absorbed by cells by endocytosis, intestinal cells or M cells (when orally administered), attacked by immune cells, filtered by the lungs, liver and spleen and excreted by kidneys. The degree of each of these processes is size-dependent, for example, nanoparticles larger than 10 nm can avoid renal clearance, while particles larger than 200 nm can be engulfed by the MPS (mononuclear phagocytic system) and accumulate in the liver and spleen [7,100]. Nanoparticle size, as well as the molecular weight and DDA of chitosan, has also been reported to influence the degree of in-vitro cytotoxicity [96]. Usually, control of the size of nanoparticles is limited by the method of preparation and tends to be larger than desired. Particles can also be too small as they are unstable under physiological conditions [101].

Although most nanoparticles are produced in a spherical shape, it should be mentioned that shape can strongly influence the performance of the drug delivery system. Shape can affect the extent of removal by macrophages, adhesion, internalisation and in vivo distribution [100,101,102].

Another important consideration is the surface of the nanoparticle. Hydrophilic surfaces are less prone to attack by macrophages. Often, nano delivery systems have PEG attached to nanoparticle surfaces to give ‘stealth’ particles that can evade attack by the MPS. However, for chitosan nanoparticles, there is a reduction in mucus adsorption due to stearic hindrance, making them unsuitable for mucus administration [7]. In contrast, hydrophobic surfaces can have improved penetration properties. A combination of hydrophobic and hydrophilic parts, in the right proportions, on the surface may give adequate penetration properties, together with good stealth [7]. Surface charge can affect the adsorption of opsonins, which attract macrophages [103]. Sometimes, engulfment by macrophages is desirable when targeting the immune system.

As to chitosan itself, chitosan nanoparticles can be modified to optimise desirable properties according to the application, such as increased time in circulation, the breakdown and release of the API and the targeting of nanoparticles to the desired site of action. For example, penetration seems to work best for high molecular mass and a high degree of deacetylation of chitosan [7]. A coating of PEG may prevent an attack by MPS. A combination of hydrophobic and hydrophilic parts, in the right proportions, on the nanoparticle surface may both prevent attacks by the MPS and give good penetration properties. A larger charge on the chitosan nanoparticle can increase its carrying ability and can serve as a pH-dependent drug carrier. Good design assumes a good knowledge of the influences on nanoparticle properties. As previously mentioned, the size and shape of nanoparticles are important. Jesus et al. found that larger-sized nanoparticles, made using the ionic gelation method with TPP, were favoured when the chitosan was of higher molecular weight and DDA [96]. Chitosan nanoparticles can, in principle, also be engineered for active targeting by attaching ligands, for example, antibodies, to the surface.

Some advantages of chitosan nanoparticles are:Less toxicity compared with other materials used for making nanoparticles;Enhanced biocompatibility;Has a mucoadhesive character;Stability;Can be used to deliver a wide variety of drugs;Can be produced using very mild conditions;Site-specific drug targeting;Increased therapeutic index of the drug;Frequent, expensive, and unpleasant dosing is reduced;Can be engineered to reduce the drug in a controlled way.

Additionally, some disadvantages of chitosan nanoparticles are:Low mechanical resistance;Difficult to control pore size;Possible contraction;Difficult electrospinning for pure chitosan;Preparation by cross-linking can affect the intrinsic properties of chitosan;Low solubility in neutral and alkaline pH;Method of preparation depends on the drug to be delivered [7].

Some of the above limitations of chitosan nanoparticles may be reduced or eliminated using nanoparticles made of chitosan derivatives or other materials added to the chitosan [7]. However, as previously mentioned, such modifications can come at the cost of complexity and reproducibility [72].

## 5. Studies of Chitosan Nanoparticles Carrying Quercetin

Recent literature on chitosan nanoparticles as a potential quercetin delivery system for some specific diseases is given in Table 2. A literature search for other diseases potentially treatable by quercetin, such as diabetes, Alzheimer’s and cardiovascular disease, only gave results for studies with either non-chitosan nanoparticles or another API.

### 5.1. Cancer Treatment

As previously mentioned, quercetin can affect signalling pathways that can potentially work against cancer by modulating cell apoptosis, preventing migration and tumour growth [64]. Some of these changes in signalling molecule concentrations are given in Table 1.

Quercetin downregulates P13K/Akt, NFkB extrinsic pathways and Bcl-2 intrinsic pathways while upregulating the FasL extrinsic pathway and the p53, Bax, FasL, p38 MAPK intrinsic pathway, leading to more apoptosis of unhealthy cells. Downregulation of MMP-9 discourages the migration of cells, helping to prevent metastasis and the inhibition of the P13K/Akt, EGFR/Akt, Her/neu pathways. NFkB reduces the growth of tumours, and an increase in p53 and p21 causes cell cycle arrest [64].

Regarding quercetin-loaded chitosan-based nanoparticles, the following studies have been made.

#### 5.1.1. Breast Cancer Treatment

De Oliveira Pedro et al. used amphiphilic chitosan nanoparticles to study the possibility of treating breast cancer with quercetin delivered by these nanoparticles [104]. Chitosan was made amphiphilic by grafting (5-bromopentyl) trimethylammonium bromide (BPTA) and dodecyl aldehyde (DDAld) onto chitosan. BPTA gave the hydrophilic groups and DDAld gave the hydrophobic groups of the resulting amphiphilic molecules. The advantage of this modification was to make the delivery system pH-sensitive. Nanoparticles were produced by self-assembly. Two types of nanoparticles were made, with differing degrees of BPTA and DDAld grafted on. These were labelled CHP_40_D_5_ and CHP_40_D_30_, where P stands for BPTA, D stands for DDAld and the subscripts give the approximate degree of substitution percentage. The exact degrees of substitution were 42.2%, 5.2% and 33.8%, which were determined by ^1^H NMR spectroscopy.

The performance of the delivery system was studied using several methods, including in-vitro drug release, the MTT (3-(4,5-dimethylthiazol-2-yl)-2,5-diphenyltetrazolium bromide) assay on breast cancer cells MCF-7, cellular uptake and blood compatibility.

In-vitro drug release studies showed that quercetin release was higher at pH 5.0 than at pH 7.4. This suggested better targeting towards tumours that have acidic microenvironments. The Korsmeyer–Peppas model was fitted to experimental data and suggested that the predominant release mechanisms were Fickian diffusion at pH 7.4 but anomalous diffusion at pH 5.0.

The MTT assay showed that unloaded nanoparticles were non-toxic to cultivated breast cancer cells but that quercetin-loaded CHP_40_D_5_ nanoparticles were more cytotoxic to these cancer cells than free quercetin (despite having released only a fraction of quercetin during the incubation period). The MTT assay showed less efficient cell killing for CHP_40_D_30_, perhaps because the quercetin is less readily released from the more numerous hydrophobic groups on the chitosan backbone.

Cellular uptake was evidenced by fusing a chitosan-affinity superfolder green fluorescent protein (CAP-sfGFP) with the chitosan derivative. The cellular distribution of nanoparticles was imaged using confocal laser scanning microscopy. The results suggested that some nanoparticles accumulated on cell membranes while others penetrated the cells.

Blood compatibility tests showed that the nanoparticles were compatible with blood.

The size of nanoparticles was determined using DLS (dynamic light scattering) with non-invasive backscattering (DLS-NIBS). Particle size ranges were 235 to 312 nm for unloaded nanoparticles and 490 to 502 nm for quercetin-loaded nanoparticles. UV–vis spectroscopy was used to determine encapsulation efficiencies (EEs), which were good at pH 5.0, being 71% for CHP_40_D_5_ and 83% for CHP_40_D_30_. For pH 7.4, the corresponding values were only 18% and 21%. Thus, self-assembly should be performed at pH 5.0, and quercetin attachment is favoured by the presence of more hydrophobic sites on the chitosan backbone. However, because the MTT assay showed less efficient cell killing for CHP_40_D_30_, this variety of nanoparticle may not necessarily be the better one.

Another study of a possible quercetin delivery system to treat breast cancer was made by Elsayed et al. [105]. They, in fact, used CuO nanoparticles that were coated with chitosan, and then, quercetin was conjugated to the chitosan surface. When loaded with quercetin, these NPs will be referred to as CuO-ChNPs-Q. CuO NPs can be good at destroying cancer cells through oxidative stress but may also be toxic to healthy cells. It was hoped that a coating of chitosan with quercetin on the surface would provide cytotoxic properties to cancer cells but not to healthy cells. This hope was realized, according to this study.

CuO-ChNPs-Q nanoparticles were characterised by the usual methods: DLS (dynamic light scattering), TEM (transmission electron microscopy), FTIR (Fourier transform infrared spectroscopy) and UV–vis (ultraviolet and visible spectroscopy). TEM gave a size of loaded coated particles of about 50 nm, with a shield of approximately 5–6 nm, confirming the conjugation of quercetin to the surface. Zeta potential was −17.6 mV. These values can be compared with CuO NPs of about 26 nm in size and zeta potential of –24.4 mV. DLS gave hydrodynamic sizes of 236 and 319 nm for CuO NPs and quercetin-loaded chitosan-coated NPs, respectively. PDIs (polydispersity index) were low at 0.21 and 0.25, respectively, showing a narrow size range. UV–vis spectroscopy was used to determine encapsulation efficiency (EE) and was found to be about 72%.

In-vitro release studies showed slow release in the first three hours, with quicker release after this time. After about 12 h, all quercetin had been released.

Toxicity to cells was assessed using the MTT assay. In-vitro studies were performed with three human cancer cell lines and one normal cell line: mammary gland cancer cells (MCF-7), colorectal adenocarcinoma (CaCo-2), hepatocellular carcinoma (HepG-2) and normal lung fibroblast (WI38). IC50 values were determined and compared with doxorubicin as the standard anti-cancer drug. The IC50 values are given in Table 3.

For the healthy WI38 cells, the low doxorubicin IC50 value of about 7 μg/mL shows that doxorubicin is clearly cytotoxic. CuO NPs have an IC50 value of about 90 μg/mL, but this is higher than for the cancer cell lines. Free quercetin has the highest IC50 value, showing non-toxicity. Encapsulated quercetin in CuO-ChNPs-Q can also be considered non-cytotoxic, but the IC50 value is less than half that of free quercetin. However, studies on other healthy cell lines need to be made before drawing a conclusion on the safety of quercetin-loaded chitosan-coated CuO NPs.

The doxorubicin IC50 value of about 12 μg/mL for the Caco-2 cancer cell lines shows that these cells have the highest resistance to doxorubicin, even compared with healthy WI38 cells. IC50 values for CuO NPs and free and encapsulated quercetin are similar, of the order of about 54–70 μg/mL, suggesting similar cytotoxicity. MCF-7 and HepG-2 cells show that both CuO NPs and quercetin-loaded chitosan-coated NPs are somewhat cytotoxic, whereas free quercetin is much less cytotoxic. CuO-ChNPs-Q is most cytotoxic towards HepG-2 cells, suggesting that they perhaps have the potential to treat liver cancer better than breast cancer.

In-vivo studies were performed on Sprague–Dawley rats, which could be given tumours by treatment with 7,12-dimethylbenzanthracene (DMBA). In-vivo studies focused on the tumour size of DMBA-induced rats showed that there was a significant reduction in tumour mass for rats treated with quercetin, CuO NPs and CuO-ChNPs-Q, with the best reduction rate when using the encapsulated quercetin. The tumour weights were 22.08, 3.18, 2.23 and 0.61 g for untreated, free-quercetin-treated, CuO-treated and CuO-ChNPs-Q, respectively. The calculated tumour inhibition rate at the end of the 16-day study was 59.95% (free quercetin), 86.25% (CuO NPs) and 96.79% (CuO-ChNPs-Q). These results show that the new CuO-ChNPs-Q formulation gave the best anti-tumour ability.

Many other investigations were made, including p53 determination, semi-quantitative determination of the cytochrome C gene, histopathology and immunohistochemistry of PCNA (proliferating cell nuclear antigen) and caspase-3. p53 levels were determined by the p53 marker. Significant increases in p53 were found for rats treated with CuO NPs, free quercetin and encapsulated quercetin, with the highest increase in the latter, compared with the DMBA control group.

Possible kidney toxicity was examined by determining changes in urea and creatinine. There were no changes in urea and creatinine levels, suggesting no kidney damage.

Liver toxicity was investigated from changes in SGOT (serum glutamic oxaloacetic transaminase) and SGPT (serum glutamic pyruvic transaminase). DMBA-induced rats showed increased levels of SGOT and SGPT, suggesting liver damage. Studies of SGOT and SGPT values for various treatments on healthy and DMBA-induced rats were performed. Healthy rats showed increases in SGOT and SGPT for all treatments. DMBA-induced rats showed decreases in SGOT and SGPT for free quercetin and CuO-ChNPs-Q-encapsulated quercetin relative to control, whereas CuO NP treatment raised values relative to control. This suggests that CuO NPs were the most toxic, but free quercetin and encapsulated quercetin in CuO-ChNPs-Q gave the least toxicity to liver function in DMBA-induced rats.

Induction of apoptosis by treatment with CuO, free quercetin and CuO-ChNPs-Q was investigated using annexin V and PI (propidium iodide) assays. The results suggested a strongly significant increase in apoptosis for all treatments on DMBA-induced rats relative to control. The free quercetin and CuO NP treatments gave similar apoptotic cell percentages. The CuO-ChNPs-Q treatment gave the highest apoptotic cell percentage.

Cell cycle arrest during cell division was also investigated. The results showed deregulation in DMBA-induced rats and the percentage of cells in the G2/M phase was about 70%, which was significantly higher than for healthy rats. Treated DMBA rats showed a reduction of cells in the G2/M phase, the percentages being about 12%, 11% and 1.5% for CuO NPs, free quercetin and CuO ChNPs-Q, respectively. Cells in the G0/G1 phase were determined to be about 66%, 58% and 67% for CuO NPs, free quercetin and CuO ChNPs-Q, respectively. These high values suggest cell arrest at this stage of the cell cycle and, therefore, the inhibition of cancer growth. Other studies have also shown that CuO NPs can induce cell cycle arrest at the G1/G0 or G2 phases [112,113].

Levels of p53 expression were investigated. p53 can enhance apoptosis in mutated cells, arrest the cell cycle at the G1/S phase and encourage DNA repair if the cell is not too damaged. It, therefore, prevents cancer cell proliferation. It also downregulates the Bcl-2 protein. Mutation of the gene coding for the p53 protein is one cause of cancer. Correct regulation of the p53 gene is also important. CuO NPs have been confirmed to regulate the p53 gene in a desirable way [114]. Elsayed et al. used a p53 marker to determine p53 levels in their various experimental groups of rats. They found no significant changes in the p53 marker in all the control groups compared with DMBA-treated rats. However, there were significant increases in CuO NPs, free quercetin and CuO-ChNPs-Q. This suggests that one mechanism of tumour growth inhibition is that the formulations induce more p53, which inhibits the proliferation of cancer cells by activating apoptosis and cell cycle arrest at the G0/G1 phase.

Further investigation of the induction of apoptosis was done by determining cytochrome C, Bcl-2 and caspase-3 and PCNA. In particular, caspase-3 activation is a sensitive assay for determining cell death via the apoptotic pathway [115]. Caspase-3 induces apoptosis and could be indirectly activated by Bcl-2. Bcl-2 can penetrate the mitochondrial membrane, enhancing the liberation of cytochrome C, which is one of the steps in the apoptosis signalling pathway. Elsayed et al. found decreases in cytochrome C in DMBA rats compared to control. Treatments with CuO NPs, free quercetin and CuO-ChNPs-Q produced significant increases in cytochrome C.

PCNA is expected to be high in cancer cells, and this has been confirmed for breast cancer cells [116]. The results of Elsayed et al. are summarised in Table 4.

Histological examination of the mammary glands of DMBA-induced rats showed marked anaplastic changes, with severe atypia, pleomorphism, large tumour syncytial cells, etc. Free-quercetin-, CuO NP- and CuO-ChNPs-Q-treated DMBA-induced rats showed increased numbers of neoplastic cells that had undergone necrosis and apoptosis.

#### 5.1.2. Lung Cancer Resistance

Wang et al. investigated a formulation designed to deliver quercetin as a means of sensitizing lung cancer cells to paclitaxel [106]. Non-small cell lung cancers often develop resistance to the first-line chemotherapeutic drug paclitaxel. This resistance is associated with the PI3k/Akt/mTOR and ERK pathways. Since quercetin can inhibit Akt, it is believed that it can act against resistance development.

Wang et al. developed a formulation involving chitosan nanoparticles made using the ionic gelation method with TPP and the targeting antibody cetuximab. The conjugation efficiency of the targeting antibodies was determined to be about 29%. These nanoparticles could carry quercetin or paclitaxel, but not both APIs, in a single nanoparticle. The formulation was tested on non-resistant A549 cells and resistant A549/taxol cells [90].

Nanoparticles were characterised by DLS, TEM, FTIR, DSC (differential scanning calorimetry), TGA (thermogravimetric analysis) and XRD (X-ray diffraction). DL (drug loading efficiency) and EE (encapsulation efficiency) were determined using HPLC (high-performance liquid chromatography). The quercetin-loaded nanoparticles were about 290 nm in diameter, with a PDI of about 0.3. TEM also showed uniform size distribution. Zeta potentials were about 24 mV, DL about 11% and EE about 88%. The paclitaxel-loaded nanoparticles were about 298 nm in diameter, with PDI again about 0.3, zeta potential about 22 mV, DL about 13% and EE about 92%. FTIR suggested that the APIs were in an amorphous state when in nanoparticles, without chemical reactions in the process of generating NPs.

A number of techniques and assays were used. They used the cell viability assay (MTT), cytoskeleton staining (F-actin), cellular uptake using fluorescence-labelled nanoparticles and flow cytometry to analyze apoptosis.

Results showed that the combination treatment of quercetin and paclitaxel could synergistically promote the apoptosis of A549 and A549/taxol cells. The mechanism was investigated by examining the mitochondrial membrane potential, which was inhibited in the A549/Taxol cells only by treatment with quercetin and paclitaxel in combination. Additionally, the two APIs, in combination, would damage cytoskeleton structures, as evidenced by irregular F-actin distributions.

Paclitaxel alone was shown to reduce the expression of the phosphorylation of Akt and ERK in A549 cells, while in A549/Taxol cells, a reduction in the expression of the phosphorylation of Akt and ERK was only observed when treated with both paclitaxel and quercetin. This again supports the view that quercetin can reverse resistance to treatment by paclitaxel in A549/Taxol cells.

In-vitro release studies showed that quercetin and paclitaxel were released more at pH 5.5 (quercetin 86%, paclitaxel 78% after 48 h) than at pH 7.4 (quercetin 77%, paclitaxel 75% after 48 h), slightly favouring release in cancer cell environments. Release was almost complete after 8 h.

BALB/c nude mice were inoculated with a suspension of A549/Taxol cells, and tumour growth with various treatments was reported. The results showed the smallest tumour growth when treated with both quercetin and paclitaxel in a nanoparticle formulation.

#### 5.1.3. Liver Cancer

Lui et al. used chitosan-based NPs to deliver quercetin with a known cancer therapy drug, doxorubicin [107]. It is believed that there is a synergistic effect between the two APIs [117,118]. The co-delivery system was aimed towards liver cancer. The loaded NPs had a particle size of 160–180 nm and a zeta potential of about 28 mV and were sub-spherical in shape [107]. Making the NPs was a complex procedure, and the NPs were designed to have a pH and redox responsiveness. Some of the initial steps involved making PEG-modified trimethyl chitosan via a Schiff-base reaction, which gave the pH sensitivity.

Doxorubicin was conjugated with a substance with a disulphide bond via disulphide bond linkage, giving redox sensitivity. NPs were produced by self-assembly. Quercetin was encapsulated by these NPs after sonication. It was shown that the PEG protected NPs from plasma protein adsorption (which might lead to the aggregation of NPs) at normal blood pH, but the PEG was removed in acidic tumour microenvironments. The encapsulated APIs are also rapidly released in tumour environments due to the breakage of disulphide bonds [107].

The formulation was tested in-vitro for drug release. In-vivo experiments with mice that were inoculated with hepatoma showed the ability of the intravenously injected formulation to inhibit tumour growth compared with control.

#### 5.1.4. Colorectal Cancer

Another study using co-delivery was undertaken by Patil and Killedar. They studied the co-delivery of quercetin with gallic acid, with the potential to treat colorectal cancer [108]. The sources of quercetin and gallic acid were pomegranate and amla fruit, respectively. There was partial extraction using Soxhlet extraction with ethyl acetate. Extracts were shown to be more potent than non-extracts. The characterisation of extracts was made using UV–vis spectroscopy, mass spectroscopy and HPTLC (high-performance thin-layer chromatography).

The formulation was made using an o/w nanoemulsion of gallic acid and quercetin dissolved in molten glyceryl monooleate (GMO); then, poloxamer 407 was added and sonicated; then, chitosan added dropwise, sonicated and then subjected to high-pressure homogenisation. Nanoemulsion was finally lyophilised with mannitol as a cryoprotectant.

Nanoparticles were characterised by DLS, XRD, SEM (scanning electron microscopy) and DSC. Hydrodynamic mean diameters were about 214 nm, and the zeta potential was 28 mV. XRD confirmed that quercetin and gallic acid were in an amorphous state in NPs.

In-vitro release studies were made in PBS (phosphate buffer solution) at pHs 2.0, 4.5, 6.8 and 7.4 in dialysis membrane sacs to stimulate the ileo-colon. It was shown that the release of gallic acid was about 78% and quercetin about 79% after 24 h, with negligible burst release.

Loading efficiencies and entrapment efficiencies were determined by dissolving lyophilised NPs in DMSO (dimethylsulfoxide), centrifuging and the supernatant examined using a UV–vis spectrometer. Loading efficiencies were found to be about 3% for quercetin and gallic acid; entrapment efficiencies were about 77% for gallic acid and 79% for quercetin.

In-vitro toxicity on human colorectal cell lines (HCR 116) was studied using the MTT assay. There was a comparison with cisplatin as a standard. IC50 values were found to be 60.32 μg/mL for polyherbal extracts and 36.17 μg/mL for unloaded chitosan NPs, while for the quercetin/gallic acid loaded NPs, it was as low as 10.55 μg/mL.

In-vivo studies were performed on Wister rats. Colorectal cancer was induced by intraperitoneal injection with DMH (1,2-dimethyl-hydrazine). The studies revealed a reduction in antioxidant enzymes GSH (glutathione), SOD (superoxide dismutase) and CAT (catalase) in DMH-treated rats. Quercetin- and gallic-acid-treated rats showed changes in RBCs (red blood cells), WBCs (white blood cells), lymphocytes, and neutrophils compared with the DMH group without treatment.

Colonic tissue sections revealed a shrinkage in epithelial linings, distorted nuclei and obtrusive adenoma with DMH rats but were nearly normal for rats treated with the quercetin/gallic acid nanodelivery system.

### 5.2. Dermal Administration for Skin Problems

Although quercetin is lipophilic and has a molecular mass of less than 500 Da, it does not easily cross the stratum corneum, the uppermost layer of the skin. This is attributed to the presence of polar hydroxyl groups [119]. However, Nan et al. showed that quercetin could more easily penetrate the skin of mice when carried by TPP-chitosan nanoparticles [39]. This was accounted for by the positive charge on the chitosan nanoparticles as the skin cell cytomembrane is negatively charged [120]. Nan et al. also suggested a disruption of the stratum corneum by the loaded nanoparticles breaking the close conjugation of the corneocyte layers [1,121].

The pH of the skin is about 5.4 to 5.9 [122], which may favour the release of quercetin from the nanoparticles. The hydrophobic nature of the released quercetin suggests retention in the epidermis as it cannot dissolve and enter the systemic circulation.

The antimicrobial properties of chitosan also make it a good vehicle for dermal applications.

#### 5.2.1. UVB Protection

Nan et al. suggested that quercetin-loaded nanoparticles could treat inflammation due to exposure to UVB light [1]. They carried out in-vivo experiments on C57BL/6 mice models and in-vitro experiments on human skin cell cultures (HaCaT cells). Their results showed that dermally administered quercetin inhibited the UVB-induced NF-kB/COX-2 signalling pathway. COX-2 (cyclooxygenase-2) is an enzyme which converts arachidonic acid into prostaglandin PGE2, which increases vascular permeability and promotes edema [123]. Thus, quercetin suppresses the production of prostaglandin PGE2, inhibiting the inflammation and edema of skin, which normally occurs after exposure to UVB light. They also reported that quercetin-loaded nanoparticles protected HaCaT cells from dying after exposure to UVB light. While cell death can be desirable in preventing cells from mutating to cancerous cells, in view of the anti-cancer properties of quercetin, the treatment may be considered beneficial.

#### 5.2.2. Wound Healing

Quercetin has promising potential for wound healing due to its antioxidant, anti-inflammation, collagen deposition, angiogenesis, and fibroblast proliferation properties [45]. It is reported that 0.3% of quercetin applied topically significantly speeds up wound healing [45]. Quercetin acts on signalling pathways by upregulating transforming growth factor (TGF)-β, CD31, vascular endothelial growth factor (VEGF), CD31, IL-10, α-SMA, GAP-43 and PCNA [45], downregulating TNF-α [124] and inhibiting the MAPK pathway [125].

Choudhary et al. used quercetin-loaded chitosan-TPP nanoparticles to study possible dermally applied delivery systems to treat wounds using the Wister rat animal model [93]. They reported that chitosan-TPP nanoparticles loaded with quercetin (0.03%) resulted in a marked reduction in tumour necrosis factor-alpha (TNF-α) and an increase in IL-10, VEGF and (TGF)-β. Better healing quality was evidenced by increased blood vessel density, decreased inflammatory cells, an increased number of myofibroblasts, the deposition and arrangement of collagen fibres and re-epithelialisation [109].

### 5.3. Ocular Diseases

Quercetin as an antioxidant has the potential to treat disorders of the eye. Lan et al. investigated quercetin-loaded chitosan nanoparticles to prevent cataracts [110]. Their studies included the use of Wistar rat pups and New Zealand albino rabbits and the physical–chemical techniques of FTIR and PXRD (powder X-ray diffraction analysis). They used Coumarin6 as a fluorescent marker with fluorescence microscopy to investigate drug distribution in the eyes of the animal models.

The study investigated a formulation based on the inclusion complex hydroxylpropyl beta cyclodextrin (HP-β-CD), which carries quercetin. This inclusion complex alone had poor corneal adhesion, so it was coated with a chitosan-N-acetyl-L-cysteine conjugate (CS-NAC). The chitosan-N-acetyl-L-cysteine conjugate gives strong corneal adhesion and suitable viscosity. The thiol groups in the CS-NAC helped form disulphide bonds with mucin on the surface of the cornea. The chitosan part of the conjugate gave good biocompatibility and perhaps helped penetration through the cornea due to its tight junction opening properties. It was found that the optimal quantity of HP-β-CD was 20% because this gave the highest accumulative amount of quercetin within 3 h after administration. The optimal quantity of CS-NAC was found to be 0.1%, being high enough to provide adhesive force to the cornea but not so high as to prevent quercetin from being released from the inclusion complex and permeating through the cornea. The formulation was found to be non-irritating, with the hydration level within a safe range.

The formulation was topically applied to the cornea surface. Permeability studies using Franz diffusion cells showed permeation into the cornea, aqueous humour, iris-ciliary body, lens and vitrous body. The main target, in order to prevent cataracts, is the lens. After 60 min, the concentration of quercetin was about 90 ng per g of quercetin administered. This value is small compared with the 2000–3000 ng found in the cornea and the iris-cilary body. However, the permeability was much better than using the inclusion complex alone without the CS-NAC coating. The authors suggest two mechanisms for the improved permeability of quercetin: firstly, the coating with CS-NAC gives a hydrophilic shell that can penetrate the hydrophilous mucus layer on the surface of the cornea, and secondly, liposoluble substances such as cholesterol and phospholipids may be extracted from cell membranes, breaking down the barriers, and letting the quercetin penetrate the cornea [94].

### 5.4. General Antioxidant

Moon et al. report a detailed study of a delivery system involving chitosan in NPs carrying quercetin as a general antioxidant that could be administered orally, in liquid form, as a preventive medicine or supplement [111]. Quercetin-loaded nanoparticles were made using a pH-driven encapsulation method. However, the main ingredient of the NP formulation was soluble soybean polysaccharide (SSPS), a negatively charged water-soluble natural polymer, a by-product of tofu or soymilk production, with excellent film-forming ability and strong adhesive, antioxidant and emulsifying properties. The smaller amount of chitosan in the formulation was mainly to make the nanoparticles more stable.

The nanoparticles were characterised by DLS, FTIR, DSC and TEM. The size of the nanoparticles was found to be quite small, 20.24 and 24.44 nm in radius, when unloaded and loaded. PDI was better when loaded, being 0.59, but was 0.80 when unloaded. Zeta potentials were −17.5 and −24.5 mV for unloaded and loaded NPs; the negative values were due to the predominance of SSPS in the formulation. DSC confirmed the loss of the crystalline structure of quercetin when encapsulated. Encapsulation efficiency was generally very high at 98% for the concentrations 0.25, 0.5 and 1.0 mg/mL of quercetin, but it decreased to 79.9% for the higher concentration of 2.0 mg/mL, probably due to the supersaturation of quercetin at this higher concentration. Loading capacities were from 12.32 to 76.79 mg (quercetin)/g (SSPS).

Although the study was focused mainly on the general antioxidant effects of the formulation, some potential anti-cancer in-vitro studies were performed on cell lines HCT-116 (human colorectal cancer), U20S (human osteosarcoma) and CCD-986sk (human fibroblasts). Anti-proliferation activity of free quercetin (dissolved in DMSO), free quercetin in media and nanoparticle-formulated quercetin (in media) was assessed using the MTS assay (3-(4,5-dimethylthiazol-2-yl)-5-(3-carboxymethoxyphenyl)-2-(4-sulfophenyl)-2H-tetrazolium) on the HCT-116 and U2OS lines during a 72 h period. It was found that free quercetin in DMSO was most effective in inhibiting cell proliferation, followed by quercetin in nanoparticle formulation. Free quercetin in media was not effective due to poor solubility. Comparison of free quercetin in DMSO and in media on CCD-986sk cells also confirmed the importance of the dissolving ability of free quercetin.

Annexin V assay was performed to study the possible promotion of intrinsic and extrinsic pathways to apoptosis. Chitosan and SSPS treatment of HCT-116 cells showed values of 2.40% for early apoptotic cells and 0.26% for late apoptotic cells. In contrast, quercetin-treated cells showed higher values of 28.4% and 1.17% for early and late apoptotic cells. Furthermore, for quercetin-treated cells, the pro-apoptotic protein expression levels of NAG-1 and p53 were found to be increased, while anti-apoptotic protein SP1 levels were decreased, confirming the promotion of apoptosis.

DPPH (2,2-diphenyl-1-picrylhydrazyl) and ABTS (2,2’-azino-bis(3-ethylbenzothiazoline-6-sulfonic acid) antioxidant assays were performed on free and encapsulated quercetin and compared with the ascorbic acid standard. The results for the ABTS assay gave results that the vitamin C equivalent antioxidant capacity was about 707 mg/g dried weight for free quercetin and 744 mg/g dried weight for formulated quercetin. Unloaded NPs were reported to have only 72 mg/g dried weight of equivalent antioxidant ability. DPPH assays showed that scavenging activities increased with the concentration of quercetin but were higher for free quercetin and consistent with the ABTS assay. IC50 values were about 77, 94 and 175 µg/mL for vitamin C, free quercetin and loaded nanoparticle formulations, respectively.

Further antioxidant activity was studied using a luciferase construct containing ARE (antioxidant response element), an NRF2 responsive reporter, transfected into HCT-116 cells. Compared with untreated cells, luciferase activity increased slightly after treatment with unloaded NPs but increased 2-fold after treatment with free quercetin or quercetin-loaded nanoparticles. This confirms an antioxidant ability due to the activation of NRF2 protein.

Anti-inflammatory activity was confirmed by the treatment of Raw264.7 mouse macrophages with free quercetin, quercetin-loaded nanoparticles and unloaded nanoparticles. After treatment, Western blot analysis showed an increase in iNOS protein expression levels for free quercetin and quercetin-loaded nanoparticles [111].

### 5.5. Other Diseases

Other diseases potentially treatable with quercetin, such as diabetes, cardiovascular disease, immunity problems, diseases causing cognitive damage and cosmetic applications, such as quercetin as a skin whitener, have not been mentioned. A literature search was performed, and there were plenty of results, but the nanoparticle delivery systems mentioned were not based on chitosan. Thus, there is still potential for new research into quercetin-carrying chitosan-based nanoparticle delivery systems to treat these diseases.

## 6. Applications of Quercetin-Loaded Chitosan Nanoparticles

As we have seen, the therapeutic applications of chitosan NPs carrying quercetin are the possible treatment for certain cancers, such as breast, colon and liver cancer; treatment against resistance to cancer drugs; skin care and sun blocking creams; treatment of certain types of pigmentation disorders, such as vitiligo; treatment of wounds; prevention of cataracts; and antioxidant supplementation. There may also be potential to treat diabetes, cardiovascular diseases, immunity problems and diseases causing cognitive damage, such as Alzheimer’s and Parkinson’s disease.

The use of chitosan provides a good way of using the waste product of shellfish in the food industry. Therefore, other potential applications of the use of chitosan and chitosan nanoparticles are also motivated by environmental concerns. Furthermore, as was mentioned in the introduction, chitosan nanoparticles can be used to remove the heavy metal pollutants produced by industries [2,3,4] or recover important materials such as silver from photographic films [5].

The applications of chitosan NPs alone are many, and a review of all the potential applications is beyond the scope of this paper. Divya et al. have written a short review of the applications of chitosan nanoparticles, which include the following:Tissue engineering;Cancer therapy;Antioxidants;Drug delivery systems;Enzyme immobilization support;Encapsulation of biologically active compounds;Water treatment;Antimicrobial agents;Agriculture [126].

Here, we will restrict our description of other applications of chitosan nanoparticles to those loaded with quercetin. Since quercetin is an antioxidant, and chitosan has antimicrobial properties, there is an obvious application to the food industry. Two studies on such potential applications are described below.

Roy and Rhim described how quercetin-loaded chitosan NPs (QCNPs) can improve the nanocomposite films used for food wrapping. As an antioxidant, quercetin helps the film to preserve the food within. Furthermore, the films with added QCNPs had excellent UV-blocking properties and significantly improved mechanical, thermal, and water vapour barrier properties. The release of quercetin from QCNP-added chitosan films was investigated using several food simulant solutions. Quercetin showed higher release under acidic and alcoholic conditions. As well as good antioxidant properties, the fabricated films also showed some antibacterial activity [127].

Valencia et al. made and studied quercetin-loaded lecithin–chitosan nanoparticles with the aim of applying them as food preservatives.

The loaded NPs were characterised with DLS, FTIR, TEM, TGA, and DSC. The size was 240 nm and zeta potential 39 mV, suggesting stability against aggregation. TEM showed the particles to be spherical. The EE was very high at 98%.

Shelf-life stability was evaluated by monitoring changes in size, PDI, zeta potential and antioxidant ability over a period of 28 days at temperatures 4 and 30 °C. The studies showed no changes in these values, suggesting the shelf-life stability of 28 days at refrigerated and room temperatures. Antioxidant strength was studied using DPPH and ABTS and showed antioxidant properties comparable with free quercetin.

Antimicrobial properties were evaluated using a number of species of Gram-negative bacteria and MIC (minimum inhibitory concentration) and MBC/MFC (minimum bactericidal/fungicidal concentration) assays. The NPs appeared to have antimicrobial activity against all the Gram-negative bacteria tested, especially against certain species. However, there appeared to be no anti-fungal activity.

Cytotoxicity was tested on L-929 (murine fibroblast cells) and PBMC (human peripheral blood mononucleated cells) cell lines using the MTT assay. The loaded and unloaded NPs were not cytotoxic to these non-neoplastic cell lines, with less anti-proliferation effect than free quercetin [128].

## 7. Conclusions

Nanoparticles show great promise for drug delivery systems that can deliver drugs with better targeting, less toxicity, less biodegradation and, hence, better bioavailability and therapeutic effect. Many APIs that were formerly considered too weak to be effective can be reassessed when delivered by nanoparticles. Consequently, there have been many studies and publications on potential drug delivery systems using nanoparticles, some for delivering weakly therapeutic agents such as phytochemicals. This can be overwhelming to review, and therefore, this review is limited to nanoparticles made from chitosan and carrying the weakly therapeutic phytochemical quercetin. This has produced a limited and manageable number of recent studies but covers a wide range of diseases to be treated. Furthermore, the methods of producing the nanoparticles are various, ranging from a CuO core coated with chitosan to modified amphiphilic chitosan.

These studies usually gave a thorough characterisation of the nanoparticles produced as well as an investigation of the release kinetics. Some of the studies investigating potential cancer therapy have been extremely thorough, investigating not only cytotoxicity on cancer and healthy cell lines but also determining changes in relevant molecular markers for liver and kidney damage, signalling molecules, the extent of apoptosis and cell cycle arrest. Effectiveness of treatment was investigated by the effect of treatment on induced tumours in animal models.

The studies reviewed here have all shown agreement that chitosan nanoparticles carrying quercetin are more effective than the administration of unencapsulated quercetin. Usually, the evidence answering the questions posed by the researcher was strong and unambiguous. Particularly interesting were studies showing the benefit of combining quercetin with a stronger and commonly used API for treatment, such as paclitaxel and doxorubicin. In vivo studies using animal models and in vitro studies using cell cultures suggest the therapeutic value of quercetin for breast cancer, lung cancer resistance to paclitaxel, liver cancer, colorectal cancer, UVB protection, wound healing, cataracts and general oxidative stress.

The studies discussed in this review have good scientific value in investigating the possibility of improving quercetin bioavailability and the effect of quercetin in treating certain disease states. Commercially, although there is the potential for future formulations of chitosan nanoparticles carrying quercetin for oral, dermal and intravenous administration, there is obviously still much more to do to produce a marketable product. While chitosan has many advantages as a biomaterial, its main disadvantage is probably its variability according to its source and processing. This makes reproducibility problematic, and the commercial use of chitosan nanoparticles may be prohibited because of the impossibility of the standardisation of the final product. Safety issues such as possible allergic reactions to chitosan nanoparticles, liver damage, and good reproducibility are also potential problems. Even with better targeting, quercetin probably cannot replace existing drugs in treating many of the diseases considered here but may be found to improve treatment when used in combination with the relevant therapeutic drugs.

Treatment with quercetin for other important diseases, such as diabetes, cardiovascular disease, immunity problems, diseases causing cognitive damage and cosmetic applications such as skin whitening, has been considered in nanoparticle formulation. However, chitosan nanoparticle delivery systems for these diseases seem to be unexplored. Thus, there is an opportunity for further research here.

## Figures and Tables

**Figure 1 ijms-24-03293-f001:**
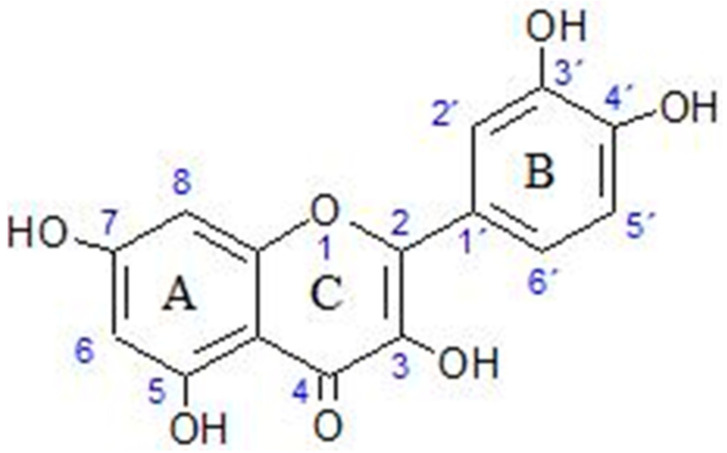
Molecular structure of quercetin.

**Figure 2 ijms-24-03293-f002:**
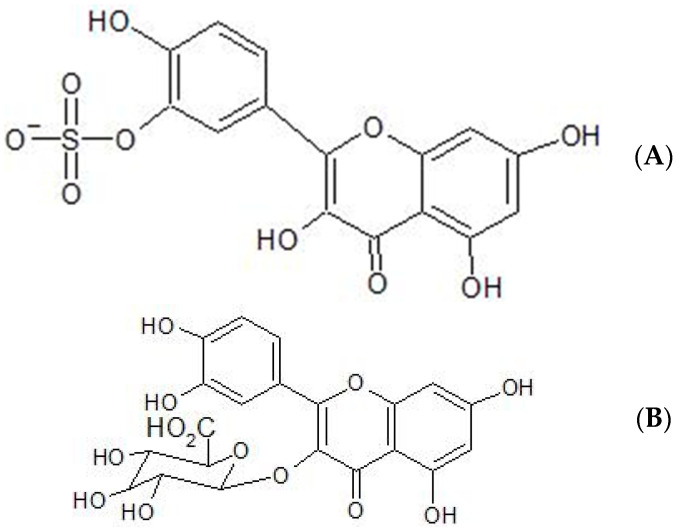
Molecular structure of (**A**) quercetin-3′-sulphate (**B**) quercetin 3-O-β-D-glucuronide.

**Table 1 ijms-24-03293-t001:** Some examples of the effects of quercetin on signalling molecules. ↑ indicates an increase, ↓ indicates a decrease. Data from references [40,45,46].

Abnormal or Disease State	Signalling Molecules or Enzyme Activity Reduced or Increased by Quercetin	Effect	Reference
Cancer	p53↑, p21↑, Caspase-3↑	Cell cycle arrest, Reduced angiogenesis and cell proliferation	[45]
	Extrinsic pathways P13K/Akt↓, NFkB↓, FasL↑ Intrinsic pathways p53↑, Bax↑, Bcl-2↓, ↑FasL↑, p38 MAPK↑	Apoptosis	
	MMP-9↓	Prevention of cell migration (metastasis)	
	p53↑, Bax↑, Bcl-2↓, ↑FasL↑, p38 MAPK↑	Reduction of tumour growth	
Acne	TLR-2↓, DNA gyrase↓	Inhibition of nucleic acid synthesis of bacteria	[45]
Psoriasis	GSH↑, SOD↑ IL1β↓, TNFα↓	Suppresses inflammation	[45]
Hyper pigmentation	Tyrosinase↓	Reduces melanogenesis	[46]
Diabetes		α-glucosidase and α-amylase	[40]

**Table 2 ijms-24-03293-t002:** Potential chitosan nanoparticle drug delivery systems for quercetin. Data from [1,104,105,106,107,108,109,110,111].

Disease Targeted	Administration Route	Type of Nanoparticle	Preparation Method	Comments	References
Breast Cancer	Intravenous injection	Hydrophilic and hydrophobic moieties attached to chitosan so that they are amphiphilic and release best at pH 5.0	Self-assembly	Amphiphilic chitosan	[104]
Breast Cancer	Intraperitoneal injection	CuO NPs coated with chitosan and TPP	CuO NPs coated with chitosan using ionic gelation with TPP	Nanoparticles contain CuO	[105]
Lung Cancer	Intraperitoneal injection	Chitosan with TPP and targeting antibody	Ionic gelation	Paclitaxel-resistant lung cancer cells	[106]
Liver Cancer	Intravenous injection	pH and redox responsive with trimethyl chitosan, disulphide bridges and PEG	Self-assembly, sonication but see original paper for details	Co-delivery of quercetin with doxorubicin, which act synergistically	[107]
Colorectal Cancer	Oral	Lipid core of glyceryl mono oleate with chitosan shell	o/w nanoemulsion with sonication, high-pressure homogenisation and lyophilisation	Co-delivery of quercetin with gallic acid	[108]
UVB damage to skin	dermal	Chitosan with TPP	Ionic gelation	In-vitro and in-vivo studies	[1]
Wound healing	dermal	Chitosan with TPP	Ionic gelation	Wister rat model	[109]
Cataract prevention	Topical application to cornea	Cyclodextrin inclusion complex coated with chitosan-*N*-acetyl-l-cysteine complex	Solvent evaporation method	Improvement of permeability into lens due to thiol groups of cysteine and perhaps chitosan. Evidence for prevention of cataract is weak	[110]
General antioxidant (cancer prevention, anti-inflammation)	Oral as liquid dosage form	Soybean polysaccharide (SSPS)/chitosan	pH-driven encapsulation method, with quercetin added to SSPS at pH 12.0 and then adjusted to pH 7 before adding chitosan	Chitosan is the minor ingredient that makes SSPS nanoparticles more stable	[111]

**Table 3 ijms-24-03293-t003:** IC50 values in µg/mL—adapted from Table 2 of Elsayed et al. [105] with permission.

	HepG-2	MCF-7	CaCO-2	WI38
CuO NPs	38.79 ± 2.8	55.65 ± 3.4	66.67 ± 3.7	93.13 ± 6.91
CuO-ChNPs_Q	26.08 ± 2.3	46.89 ± 2.9	54.29 ± 3.4	215.6 ± 24.7
Free quercetin	103.9 ± 13.7	118.55 ± 22.5	69.34 ± 4.6	454.5 ± 48.1
Doxorubicin	4.50 ± 0.2	4.17 ± 0.2	12.49 ± 1.1	6.72 ± 0.5

**Table 4 ijms-24-03293-t004:** Changes in caspase-3 antibody and PCNA antibody in mammary gland tissue. PCNA is proliferating cell nuclear antigen (adapted from Elsayed et al. [105]) with permission.

Treatment	Lining Epithelium	Tumour Mass	Neoplastic Cells
CONTROL healthy rats untreated	Slight expression of caspase-3 and PCNA		
DMBA-induced rats	Overexpression of PCNA	Slight expression of caspase-3	Low expression of caspase-3
CuO NPs on DMBA-induced rats			Increase in caspase-3 Decrease in PCNA
Free quercetin on DMBA-induced rats			Increase in caspase-3 Decrease in PCNA
CuO-ChNPs-Q on DMBA-induced rats	Marked decrease in PCNA		Marked increase in caspase-3

## Data Availability

Data sharing not applicable.

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
