# Peer review of "Improvement of Therapeutic Value of Quercetin with Chitosan Nanoparticle Delivery Systems and Potential Applications"

_ijms, 2023, doi:10.3390/ijms24043293_

Round 1
Reviewer 1 Report
The manuscript titled ‘Improvement of Therapeutic Value of Quercetin with Chitosan Nanoparticle Delivery Systems and Potential Applications’ summarized the Physicochemical properties of quercetin and chitosan nanoparticle delivery systems, as well as different kinds of chitosan nanoparticle delivery systems carrying quercetin invented for various diseases. This review is favorable to further development of quercetin. However, there are still some issues to be solved before publication.
1. The main idea of this review is not clear. Authors spend more than half of the article on the separately properties of the quercetin and the chitosan nanoparticle delivery systems, but the title of this review is about the improvement therapeutic value and the potential application, I wonder whether it is better to adjust the ratio composition of the article.
2. In line 27, authors have used ‘what is quercetin?’ as title to introduce the basic properties of quercetin, it’s better to choose written language to ensure the rigorousness of this review.
3. The example author used in line 45-line 48 to confirm ‘The metabolites of quercetin that are biologically active’ isn’t clear to support this conclusion, why not list much more intuitive evidences.
4. Images not aligned. The structures in figure 2 is not aligned, it’s better to adjust it.
5. It’s kindly of you if change the format of table 1. Honestly speaking, it seems a little bit strange.
6. In line 107, authors use the searching from Google as evidence to list the contraindications of quercetin, for what is missing the scientific rigor.
7. Before line 443, probably lose a tile called ‘breast cancer’, cause other diseases that can be cured by chitosan nanoparticle systems carrying quercetin all have a title, if so please complete.
8. Last point, the parts of ‘potential application’ is not detailed, chitosan nanoparticle systems carrying quercetin do really have a promising application in the future, I wonder whether the authors can list much more content of potential application of this.
Author Response
Dear Reviewer,
Thank You very much for Your time and thoroughness in reviewing my paper titled ‘Improvement of Therapeutic Value of Quercetin with Chitosan Nanoparticle Delivery Systems and Potential Applications.’
I will address each point You have made.
- I understand that more than half of article discusses the properties of quercetin and chitosan. However, I have tried to keep the description concise but have included a lot of relevant information. I therefore wish to keep these sections as they are.
- Replaced title with ‘Basic properties of quercetin’
- Have rewritten and added references which say which metabolites are beneficial to health
- The images have been aligned
- Replaced Table 1 with a list
- I have revised these sentences
- Added title Breast cancer treatment
- Unfortunately, I have not really had time to revise this last point.
Thank You once again for Your time and I hope You are satisfied with my revisions.
Yours faithfully,
Michael Lawson (author)

Reviewer 2 Report
Figure 1 should be improved (some bonds have unnormal lengths (C4’-O) or angles). Is the notation of the ring necessary for the discussion?
Line 31: partition coefficient (1.82) of quercetin between what kind of solvents?
Line 40: aglycone form of quercetin was not previously mentioned (explain)
Lines 46-48: what means "detected," by what? Why is it a complication?
Figure 2: add notation of derivatives to each molecule
Chapter 1.2 suggest that it should describe the metabolites of quercetin – but there is a lack of quercetin metabolism, and the products of this process
Line 91: API abbreviation should be explained when used for the first time
Line 108 – please provide these articles for further reading
Line 109-110: sentence needs revision
Line 146-147 – lack of reference
Line 244: chitin is not composed of only one monomeric unit, it is copolymer as chitosan but of the different ratios between these units
Line 249 and 261: it is generally assumed that chitosan is defined when DDA > 50% (not 30%), while below, it is called chitin
Line 250: chitosan does not possess a positive charge ("its positive charge") but can undergo protonation in given external conditions
Line 258: if the pKa of chitosan is 6.5, then in neutral solutions (of pH = 7), the degree of protonation is very low (see here: Carbohydrate Polymers 153 (2016) 501–511)
Lines 263-266: sentence needs revision
Line 268: what does "basic biomaterial" mean?
Line 270: water has pH 7, and chitosan is not soluble in water
Line 280: in comparison to what kind of polymers? This statement is overestimated
Lines 282-285: several derivatives with better solubility are also produced
Line 293: "thermally less stable" – less than what?
Line 307: in this list rather, the effects of modifications are given, then the summary of the modification types
Line 334-336: sentence needs revision
Line 355: please check the characteristic of glutaraldehyde and its toxicity
Line 411 "low" mechanical resistance
Others: Line 92 (medicinal -> medical), line 297 (groups "can not be added" but the modification can occur when the reaction is taking place with NH2 or OH), line 265 (nor -> not).
The style of literature citation should be revised as in some cases (see lines 82-83 and 85-86), the duplicates at the end of particular sentences are unnecessary. In some cases, the citation style is also incorrect (see line 95, (Vida RG, 2019), line 96 should be [35-38], etc.). In many places, this same citation is given after each separate sentence (see e.g., lines 256-267)
The writing style and some statements (e.g., 'online (provided?) quercetin supplements', effect (affect?) line 155, Table 3 and line 186: absorption (? adsorption)) need to be caerfully revised.
Author Response
Dear Reviewer,
Thank You very much for Your time and thoroughness in reviewing my paper titled ‘Improvement of Therapeutic Value of Quercetin with Chitosan Nanoparticle Delivery Systems and Potential Applications.’
I will address each point You have made.
I have improved Figure 1 but left the notation for rings because it can be useful
Line 31 have added between water and octane
Line 40 Have rewritten to avoid this term
Lines 46-48. Detected by HPLC and removed sentence saying it is a complication
Have labelled the two derivatives A and B and refer to them in the Figure description
I have added some of the details known about the metabolism of quercetin and added references
Line 91 I have explained abbreviation API the first time it is mentioned
Line 108 I did not understand exactly how to provide the articles so have not provided the articles only the web sites.
Line 109-110 I have revised this.
Line 146-147 I have corrected this.
Line 244 I have corrected this.
Line 249 and 261 I have corrected this and provided reference.
Line 250 I have corrected this.
Line 258 Corrected and provided reference.
Lines 263-266 Revised this sentence.
Line 268 Rephased to avoid this question
Line 270 corrected
Line 280 Removed this statement
Line 282-285 have rewritten to include this
Line 293 Have rewritten to avoid this question and have added more about how to improve stability
Line 307 Replace ‘A summary of …’ by ‘The effects of …’
Line 334-336 Have rewritten
Line 355 Have stated that its disadvantage is its toxicity and included a reference to its toxicity.
Line 411 Replaced ‘Less’ with ‘Low’
Line 92 Replaced ‘medicinal’ with ‘medical’
Line 297 Have rewritten so this is explicit
Line 265 Replaced ‘nor’ with ‘not’
I have revised the reference citation style so that for example have references like [87-90] and removed some of duplicates.
I have removed the word ‘online’ and replaced ‘effect’ by ‘affect’. I have replaced Table 3 with a list and replaced ‘absorption’ with adsorption’.
Thank You once again for Your time and I hope You are satisfied with my revisions.
Yours faithfully,
Michael Lawson (author)
Reviewer 3 Report
Review entitled “Improvement of Therapeutic Value of Quercetin with Chitosan Nanoparticle Delivery Systems and Potential Applications” discus the potential applications of chitosan nanoparticles loaded with quercetin. The point is attractive and the review contains promising collected data. The review needs minor revision before accepting to publish.
1. Line 14, “in vitro, ex vivo and in vivo” should be “in-vitro, ex-vivo, and in-vivo” and italic, please revise throughout the manuscript.
2. The review should be beginning by introduction section contains general information about main point, general applications, and overall importance view of the point or why select this point. I recommend to citing the following for different application of chitosan nanoparticles.
https://doi.org/10.3389/fphar.2018.00826; https://doi.org/10.1016/j.cej.2021.131775; https://doi.org/10.3390/ma14092189; https://doi.org/10.1016/j.cej.2021.133967; https://doi.org/10.1016/j.jece.2022.107939; https://doi.org/10.1016/j.ijbiomac.2022.02.173
3. Figure 2 should be labeled as A and B and defined this letter in Figure legend.
4. Data in Table 1 can be summarized in text and their presence of table is unusual.
5. Table 1 can be reorganized to contain all beneficial activities and their mechanisms of quercetin.
6. Line 95, “(Vida RG, 2019).” Should be display as number, please check and revised the reference style throughout the manuscript.
7. Line 180, “From Nan et al.” should be “according to Nan et al.”
8. Table 4, “Reference to study” should be “References”
9. I recommend compressing Table 5 with Table 2.
10. Title 5, Discussion of what?
11. The review should be containing the figures for mechanisms to increase their visibility and their citation.
Author Response
Dear Reviewer,
Thank You very much for Your time and thoroughness in reviewing my paper titled ‘Improvement of Therapeutic Value of Quercetin with Chitosan Nanoparticle Delivery Systems and Potential Applications.’
I will address each point You have made.
- I have put latin terms in italics and hyphenated those necessary
- Have added Introduction with the suggested citations
- Have labelled A and B in Figure 2 and defined in legend
- Replaced Table 1 with a list
- I have left the beneficial activities in the text and did not reorganise into a table
- Corrected this oversight
- Replaced ‘From’ with ‘According to’
- Replaced with References
- I have combined Table 5 with Table 2 which is now Table 1
- I have removed the title Discussion and included the text in the Conclusions
- I am afraid I have not had the time to include such figures of mechanisms
Thank You once again for Your time and I hope You are satisfied with my revisions.
Yours faithfully,
Michael Lawson (author)
Round 2
Reviewer 2 Report
Line 81 The partition coefficient in the water/octanol system higher than 1 suggests that quercetin is better soluble in water than in octanol – thus, why is it regarded as hydrophobic?
Line 347: Not 'absorption' but 'adsorption'
Author Response
Response to reviewer 2 (2nd round)
Dear Reviewer,
Thank You very much for Your time and thoroughness in reviewing my paper titled ‘Improvement of Therapeutic Value of Quercetin with Chitosan Nanoparticle Delivery Systems and Potential Applications.’
I have addressed the minor revisions as follows.
I have changed the reference to the partition coefficient of quercetin to ‘(octanol-water partition coefficient 1.82 ± 0.32)’
I have replaced ‘reduction in mucus absorption’ to ‘reduction in mucus adsorption’
I have also added some more text to sub-section 2.2. Derivatives and metabolites of quercetin and corrected more typing errors I have found.
Thank You once again for Your time and I hope You are satisfied with my revisions.
Yours faithfully,
Michael Lawson (author)
